# The Crossregulation Triggered by *Bacillus* Strains Is Strain-Specific and Improves Adaptation to Biotic and Abiotic Stress in Arabidopsis

**DOI:** 10.3390/plants13243565

**Published:** 2024-12-20

**Authors:** Estrella Galicia-Campos, Ana García-Villaraco Velasco, Jose Antonio Lucas, F. Javier Gutiérrez-Mañero, Beatriz Ramos-Solano

**Affiliations:** Faculty of Pharmacy, Universidad San Pablo-CEU Universities, 28668 Madrid, Spain; estrellagcamp@gmail.com (E.G.-C.); anabec.fcex@ceu.es (A.G.-V.V.); alucgar@ceu.es (J.A.L.); jgutierrez.fcex@ceu.es (F.J.G.-M.)

**Keywords:** PGPR, stress, *Pseudomonas syringae* DC3000, salinity, priming

## Abstract

Plants are sessile organisms that overcome environmental stress by activating specific metabolic pathways, leading to adaptation and survival. In addition, they recruit beneficial bacterial strains to further improve their performance. As plant-growth-promoting rhizobacteria (PGPR) are able to trigger multiple targets to improve plant fitness, finding effective isolates for this purpose is of paramount importance. This metabolic activation involves the following two stages: the priming pre-challenge with no evident changes, and the post-challenge, which is characterized by a faster and more intense response. Eight *Bacillus* strains, obtained in a previous study, were tested for their ability to improve plant growth, and to protect *Arabidopsis thaliana* plants against biotic and abiotic stress. After the 16S rRNA gene sequencing, three isolates were selected for their ability to improve growth (G7), and to protect against biotic and abiotic stress (H47, mild protection, with a similar intensity for biotic and abiotic stress; L44, the highest protection to both); moreover the expression of Non-Expresser of Protein Resistance Gene 1 (*NPR1*) and Protein resistance (*PR1*) as markers of the Salicylic Acid (SA) pathway, and lipooxygenase (*LOX2*) and plant defensin gene (*PDF1*) as markers of the Ethylene/Jasmonic Acid (Et/Ja) pathway, was determined 24 h after the stress challenge and compared to the expression in non-stressed plants. The results indicated that (i) the three strains prime *Arabidopsis* according to the more marked and faster increases in gene expression upon stress challenge, (ii) all three strains activate the SA-mediated and the Et/Ja-mediated pathways, therefore conferring a wide protection against stress, and (iii) *PR1* and *PDF1*, traditionally associated to Systemic Acquired Resistance (SAR) and Induced Systemic Resistance (ISR) protection against pathogenic stress, are also overexpressed under abiotic stress conditions. Therefore, it appears that the priming of the plant adaptive metabolism is strain-dependent, although each stress factor determines the intensity in the response of the expression of each gene; hence, the response is determined by the following three factors: the PGPR, the plant, and the stress factor.

## 1. Introduction

Stress involves a number of environmental factors that negatively affect plant fitness, including biotic and abiotic factors, such as pathogenic microorganisms, water scarcity (drought), or an excess of soil salts (salinity). The United Nations has settled the goals for the upcoming years to achieve food security under climate change conditions that are threatening crop yields. Breeding technologies to improve plant genetic endowment, and better agronomic management on water usage and fertilization, are among the strategies under development to improve crop yields under water scarcity [1]. Irrespective of these approaches, the activation of the innate plant metabolism to improve plant adaptation by beneficial bacterial strains, known as plant-growth-promoting rhizobacteria (PGPR), appears as a challenging tool to approach this goal.

Plants are sessile organisms that are genetically endowed with mechanisms to overcome stress factors in order to survive. Upon sensing, plants activate adaptive mechanisms by altering Reactive Oxygen Species (ROS) homeostasis, which triggers the complex signaling network, leading to adaptation, with a differential ROS signature for each stress situation that triggers the specific response [2]. In case of water stress (drought), the activation of the Ca^2+^ pump increases the ROS concentration, triggering the expression of oxidative-stress-sensitive genes in the following two steps: first, the transcription factors, and then the functional genes [3]. This leads to osmolyte synthesis, which allows for the maintenance of water homeostasis [4], until Abscisic Acid (ABA)- or Ethylene-mediated stomatal closure leads to growth arrest if the situation is too extreme. When the water stress is due to salinity, plants accumulate Na^+^ to absorb the available water until the ion concentration eventually becomes toxic [5]; to counteract this accumulation, Na^+^ ions are pumped out, and the specific ROS signature generated activates the hormone-mediated stress response involving Ethylene (Et), Jasmonic Acid (JA), and Abscisic Acid (ABA) [3,6].

In the case of biotic stress, pathogens trigger an ROS signature, activating respiratory burst oxidase homologue (RBOH-type) receptors [7], leading to the activation of systemic responses mediated by SA. Upon pathogen sensing, the ROS burst associated to the increase in SA leads to the activation of the transcription factor NPR1, which translocates to the nuclei, where it activates the transcription of PR genes involved in protection [8,9]; the ROS burst is then systemically transmitted both by ROS and other signals. As ROS homeostasis needs to be restored, plants are endowed with antioxidant mechanisms involving enzymes and antioxidant molecules for ROS scavenging [10]. The SA pathway is traditionally known as SAR (Systemic Acquired Resistance) [11] and is part of the plant’s immune system, which also consists of the Et/JA pathway [12]. This pathway is activated by certain beneficial strains that cause a slight increase in the ROS, which triggers the Et/Ja pathway, activating the expression of defensin-type proteins (pdfs) involved in plant protection, which is termed Induced Systemic Resistance (ISR) [12,13]. This protection takes place in the following two-step process: first, the bacterial strain triggers genetic and metabolic changes that are not detectable, a physiological situation called the priming pre-challenge state; secondly, upon stress challenge, the response is faster and more intense than in non-primed plants, with a specific primed post-challenge fingerprint [14].

Among the beneficial strains able to trigger ISR are Plant-Growth-Promoting Bacteria (PGPB), an inclusive name that refers to all beneficial bacteria irrespective of their origin, while PGPR refers to those isolated from the rhizosphere [15]. These PGPB can improve plant growth and fitness via several mechanisms, like improving the nutrient availability, the biocontrol of soilborne pathogens, or causing a systemic induction of plant adaptation to stress [16,17,18]; however, there is a certain specificity between the host and the strain for systemic induction that needs to be checked. Strains that are able to prime plants induce a broad protection to many stress factors simultaneously in a process known as “priming-induced acquired stress tolerance” [19,20]. This is a three-step process, as follows: (a) exposure to priming stimuli, namely, the PGPB or a part thereof; (b) the restoration of the optimal growth conditions and epigenetic changes (the priming pre-challenge); (c) a more efficient and stronger response upon stress challenge (the priming post-challenge) [14,21].

In view of the above, it is evident that plants integrate environmental signals through ROS, and the delivery of specific beneficial bacteria to plants appears as an interesting strategy for a systemic activation of the plant metabolism for better adaptation. The beneficial effects of the PGPR in plant growth, plant protection, and plant adaptation to stress have been widely studied, with studies reporting on the specific effects and signal transduction pathways for each case [22]. The crosstalk among these pathways have also been described, revealing the complex signaling network involved in plant adaptation [23]. Interestingly, and due to this crosstalk, the PGPR appears to be able to trigger several physiological targets simultaneously, improving plant adaptation to several stress factors [24], which is a more realistic situation than a single-stress lab experimental set-up. However, there is a certain specificity between the host and the strain for systemic induction; moreover, not all strains activate protection to different stress factors. Therefore, finding effective strains able to prime plants against many stress factors is of paramount importance for agriculture under harsh conditions.

The objective of the present study was to unravel whether the signal transduction pathways triggered by beneficial bacteria on *Arabidopsis thaliana* are stress- or strain-dependent. For this purpose, a screening of eight putative PGPR strains was performed in *A. thaliana* so to identify the effective strains that are able to trigger protection against biotic and abiotic stress, by protection assays against saline, or by pathogen challenge. Three outstanding strains in growth and protection were studied for signal transduction pathways by analyzing the expression of marker genes for the SA- or Et/JA-dependent pathways.

## 2. Results

### 2.1. Selection of PGPR: 16S rRNA Analysis and Biological Assay to Biotic and Abiotic Stress

The phylogenetic tree built with the 16S rRNA gene sequences of the eight strains revealed four different groups (Figure 1). Group 1 was formed by G7, identified as *Bacillus simplex*; group 2 was formed by the following three strains: K8, identified as *Bacillus* sp., and L24 and L44, identified as *Bacillus aryabhattai*; group 3, which contained three strains, H47, L36, and L79, identified as *Bacillus amyloliquefaciens*; group 4, which was formed by L56, identified as *Bacillus* sp.

Only strain G7 significantly increased the dry weight in the non-stressed plants, while four strains significantly decreased it (L44, L56, L79, and H47) (Table 1). Under salt stress, the controls showed a lower weight than the non-stressed controls; all of the inoculated plants under salt stress showed a higher weight than the non-inoculated controls. Interestingly, some of the strains were able to maintain similar weight values in either condition (L24, L44, G7, and H47), while the other group showed an increased weight under salt stress as compared to the inoculated and non-stressed plants (K8, L56, L79, and L36).

Figure 2 shows the protection relative to the controls. All of the isolates protected against salt stress, decreasing the wilting index as compared to the controls, except for L56 and K8. Similarly, all of the strains increased the protection against the leaf pathogen, except for G7 and K8 (Figure 2b).

The following three strains were selected: G7, based on the growth stimulation (dry weight—Table 1), and L44 and H47 for the protection against biotic and abiotic stress, with L44 being the highest and H47 being moderate (Figure 2); the genetic divergence according to the 16S rDNA sequence was also considered (Figure 1). These three strains were used in a second experiment to study the photosynthesis and signal transduction pathways.

### 2.2. Effects of Stress Challenge on Photosynthetic Performance with Selected Strains

The photosynthetic efficiency of the plants inoculated with each strain under no stress, after the salt stress challenge (Figure 3), and after the pathogen challenge (Figure 4) was evaluated. Non-stressed inoculated plants showed few alterations in the photosynthetic parameters; only G7 increased the photosynthetic efficiency of photosystem II (ϕPSII), and both G7 and H47 decreased the energy dissipation (NPQ). Under the salt stress challenge, the minimum fluorescence in dark-adapted leaves (F0) significantly increased in the non-inoculated stressed controls, while the maximum potential photosynthetic efficiency of photosystem II (Fv/Fm) and ϕPSII decreased, and the NPQ was not affected. Inoculated salt-challenged plants showed changes as compared to the stress controls; changes were strain-dependent, with the only common point being a decrease in the NPQ (Figure 3).

The pathogen challenge only modified the NPQ in the non-inoculated controls, thus decreasing its values (Figure 4). Each strain created a unique fingerprint in photosynthesis which was not necessarily identical to that of the salt stress: G7 affected all of the parameters except for Fv/Fm, L44 did not modify the parameters at all, and H47 only increased F0.

### 2.3. Signal Transduction Pathway

To determine if the signal transduction pathways involved in protection were dependent on the stress or on the strain, the following four genes indicative of the SA pathway or the JA/Et pathway were studied: the genes *NPR1* and *PR1* as markers of the SA pathway, and *LOX2* and *PDF1* as markers of the Et/Ja pathway.

Plants treated with H47 (Table 2) showed a significant 1.5-fold upregulation of *LOX2* as compared to the non-inoculated non-stressed controls, and the downregulation of *PDF1*, while the SA markers were not affected. Under the salt stress challenge, *PR1* and *PDF1* were upregulated, while *NPR1* and *LOX2* were downregulated. Under the pathogen challenge, *NPR1* and *LOX2* were upregulated.

Plants treated with L44 showed the upregulation of *LOX2* (2.5-fold) and the downregulation of *PDF1*. Under salt stress, *PDF1* was upregulated. Pathogen-challenged plants showed the upregulation of *PR1* and the downregulation of *PDF1*.

G7-treated plants showed the downregulation of *PR1* and *PDF1*. Under salt stress, the SA marker genes and *LOX2* were upregulated, while *PDF1* was downregulated (Table 2). Under the pathogen challenge, only a significant upregulation of *PR1* was detected.

## 3. Discussion

The ability of plants to recruit effective strains to improve their survival is beyond any doubt. Interestingly, the rhizosphere exhibits an enormous genetic redundancy of beneficial strains, constituting an excellent source of specialized strains able to provide the plant with a tailored metabolic activation for the best adaptation for the many varying conditions of plant growth [16,17,24,25]. However, the ability of beneficial bacteria to trigger the metabolism of different plant species is not always successful, as there is certain specificity, so the identification of a wide scope strains is of paramount importance for sustainable agriculture. Furthermore, although all strains hold a beneficial potential, not all of them trigger the same signal transduction pathways in the plant [25,26,27], which only contributes to maximizing the plant’s adaptive metabolism [24].

Among the evaluated strains, not all of them induced positive effects on plant growth, as expected from their in vitro beneficial traits like auxin or siderophore production [28], with this latter trait also being indicative of plant protection [29]. The negative effects on plant growth induced by some strains have been previously related to the pre-activation of the plant defensive metabolism to trigger a quicker and more intense response upon stress challenge; this physiological state is known as priming and usually compromises the growth due to the detour of energetic resources for defense on ISR [26,30,31]. Although the priming response was first demonstrated for a pathogen challenge, the systemic induction is valid for other stress factors, as evidenced in the present study, which is an example of trans-priming [30,32]. This phenomenon refers to the stimulation of the defensive metabolism in the following two steps involving different elements: the PGPR in first instance and an intense stress (biotic or abiotic) in the second instance [21].

Despite the energetic cost of the priming state, revealed in the lower growth (dry weight) under the influence of some strains, the energetic cost is worse upon saline stress: while the controls show a huge weight loss (60%), the PGPR-treated plants overcome even the non-stressed controls. The increase in the dry weight in the salt-stressed plants could be attributed to ion accumulation [33], but not under non-stress conditions, suggesting that other mechanisms are involved in the positive effect on growth. Interestingly, the growth modifications induced by some strains were similar in all conditions (G7, L24, L44, and H47—Table 1), which presents quite a promising feature for future applications in agricultural production, as growth modifications seem to be consistent and independent from stress factors. The increase in plant fitness is achieved, among other factors, by the modulation of light reactions to lower the energy loss associated to stress, together with the reduction in the physiological oxidative stress [34,35]. It appears that one of the best targets to improve plant adaptation to stress by PGPR is photosynthesis, as photochemical quenching (ϕPSII) is enhanced while the dissipated energy (NPQ) is lowered, especially with G7 and H47, while L44 finds other potential targets (Figure 4).

Trans-priming is key for success in field applications, as the PGPR will activate the plant metabolism to overcome any stress, contributing to enhanced yields; however, the intensity in the response depends on the PGPR strain and the host (genome–genome specificity) [36,37]. The underlying mechanism involved in trans-priming relies on the ability of strains to trigger signal transduction pathways, which are interconnected to achieve a wide protection [13]; irrespective of the mechanism, the alteration of ROS homeostasis, leading to changes in gene expression, will take place [3,38].

Based on the above, and to elucidate if the signal transduction mechanisms were strain-dependent or stress-dependent, and if they were simultaneously triggered by the strain, a battery of genes were studied [22,39,40,41].

Each strain showed a specific priming profile. Under the no-stress conditions (the priming pre-challenge), H47 and L44 induced the overexpression of *LOX2* and *PDF1*, which is consistent with an Et/JA-mediated signal transduction [12] (van Loon et al., 1998). For G7, an interesting downregulation of *NPR1* and *PDF1* was detected, suggesting that the activation of both pathways took place within the 24 h period; however, 24 h after the stress challenge (4 days after the last dose of G7), when the sampling was carried out, the depletion that follows a peak in the response was detected [42]. Upon salt stress, H47 and G7 triggered both pathways, as all four genes were affected [40,43]. However, it seems that H47 responded faster according to the overexpression of the downstream genes *PR1* and *PDF1*, while the upstream genes (*NPR1* and *LOX2*) were already showing the depletion on the response; therefore, both pathways are activated simultaneously. On the other hand, G7 showed a striking peak in the overexpression of both SA marker genes and in *LOX2*, which is consistent with the wave-like response suggested by Kollist et al. [42]; interestingly, although both pathways are triggered simultaneously, there seems to be a lag between both, which probably contributes to a better adaptation. Upon pathogen challenge, H47 triggered a similar pattern than upon salt stress, thereby activating both pathways, as revealed by the overexpression of the upstream genes *NPR1* and *PDF1*; G7 only activated the SA-mediated pathway at that time point. Finally, L44, which achieved the highest protection against the salt and pathogen challenges, seemed to activate a faster response [41], as suggested by the downregulation of the gene expression (*NPR1*, *PR1*, and *LOX2*) upon the salt stress challenge, and *PDF1* upon the pathogen challenge. Although further studies should be carried out to confirm the time for the first peak of responses, it is really interesting to evidence a fast response based on the unique priming capacity of each *Bacillus* strain studied, which confers a wide protection to stress upon demand, contributing to the best adaptation [24].

In summary, (i) all three strains trigger both pathways simultaneously; (ii) the response occurs within different time frames; (iii) changes in gene expression occur earlier than 24 h post-challenge; (iv) based on the protection levels detected and the expression of marker genes, the data presented here suggest that the response follows a wave-like pattern, as highlighted by Kollist et al. [42]. In view of these data, the response triggered by each bacterial strain is strain-dependent, although stress factors modulate the intensity of the response.

## 4. Materials and Methods

### 4.1. Plant Material

*Arabidopsis thaliana* col 0 seeds, provided by the NASC (The National Arabidopsis Stock Center, Nottingham, UK), were used. Seven-day-old pregerminated seeds were transplanted to 100 mL pots filled with 60 g of a sand/peat (12:5 *v*/*v*) mixture. The plants were kept in a Sanyo MLR-350H culture chamber throughout the experiment, with a 12 h/12 h photoperiod 5/0 light intensity, and watered to maintain the soil moisture, as requested.

### 4.2. Bacterial Strains and Inoculum Preparation

The 8 beneficial strains assayed in this study were Gram-positive sporulated bacilli [21] (L79, L56, L24, L36, G7, L44, K8, and H47). All of them were isolated from the rhizospheres of *Pinus pinea* and *P. pinaster* [44]. They were able to produce siderophores (L79, G7, and H47), auxins (L56, L24, and L44), auxins and siderophores (L36), or auxins, and could degrade 1-amincyclopropane-1-carboxylate (ACC) (K8).

The bacterial strains were kept at −80 °C in a nutrient broth amended with 20% glycerol. To prepare the inocula, the strains were plated on Plate Count Agar (PCA) and incubated for 24 h at 28 °C. Then, they were inoculated in Lysogenic Broth (LB, Condalab, Madrid, Spain) and incubated under shaking at 28 °C for 24 h. Inoculation was carried out with a solution of 1 × 10^8^ cfu (colony-forming units) mL^−1^.

### 4.3. Bacterial Strain Partial Sequencing of 16S rRNA Gene and Phylogenetic Tree

The bacterial strains were identified by the partial sequencing of the 16S rRNA gene. The bacteria were grown in Lysogenic Broth (LB, Condalab, Spain) under shaking for 24 h at 28 °C. The extraction of the DNA was performed with 1.8 mL of the microbial DNA extraction kit UltraClean (Mo Bio, Carlsbad, CA, USA). The quantification of the DNA and the quality check were performed with the Nano Drop 2000 (Thermo Scientific, Waltham, MA, USA).

The universal primers for the 16S rRNA gene were as follows: 1492R and 27F [45]. The amplification reactions were carried out as follows: 5 μL of DNA (20 ng μL^−1^), 1 unit of DNA polymerase Hotsplit (Biotools, Madrid, Spain), 0.5 μL of Primer F (30 μM) and 0.5 μL of Primer R (30 μM), 2.5 μL of reaction buffer 10× MgCl_2_ (Biotools), 0.625 μL of dNTP (10 mM each, Biotools), 0.375 μL of 100% DMSO (Dimethyl sulfoxide), and ultrapure water to a final volume of 25 μL. The PCR reactions were carried out on a Gene Amp PCR system 2700 (Applied Biosystems, South San Francisco, CA, USA) at 94 °C for 2 min, followed by 10 cycles (94 °C for 0.3 min, 50 °C for 0.30 min, and 72 °C for 1 min) and another 20 cycles (94 °C for 0.3 min, 50 °C for 0.30 min, and 72 °C for 1 min and 5 s); then, the mixtures were incubated at 72 °C for 7 min. The PCR products were cleaned with the ADN UltraClean PCR Clean-up kit (MO BIO, Carlsbad, CA, USA) and sequenced on an ABI PRIMS” 377 DNA sequencing device (Applied Biosystems, Foster City, CA, USA). The software Sequence Scanner v1.0. (Applied Biosystems, Foster City, CA, USA) was used to visualize the sequences, and editing was carried out with the Clone Manager Professional Suite v6.0 software (Sci-Ed Software, Cary, NC, USA). The sequences were aligned with MAFFT v6.0 (https://mafft.cbrc.jp/alignment/software/), and annotated with BLASTN 2.2.6 (NCBI: https://www.ncbi.nlm.nih.gov/) and Ribosomal Database Project Release 10 (RDP: http://rdp.cme.msu.edu/).

To study the phylogenetic relationships among all of the strains, a rooted tree was constructed with MAFFT v7 (24 April). The evolutionary distance was inferred with the Neighbor-Joining method. A bootstrap consensus tree was inferred after 1000 repetitions to represent the evolutionary history of the analyzed taxa.

### 4.4. Experimental Set-Up

Two experiments were performed, one to select the most effective strains triggering plant protection to biotic and abiotic stress, and the second with the 3 selected strains so to unravel the signal transduction pathways involved in the protection.

#### 4.4.1. Screening for the Most Effective Isolates

Two-week-pregerminated A. thaliana col 0 seeds were transplanted to 12-well plates filled with 3:1 (*v*/*v*) peat/sand. The plants were grown in a Sanyo MLR-350H growth chamber for 9 h/15 h (350/0 μE/s m^2^) at 24 °C/20 °C, keeping a 70% relative humidity. The plants were watered twice a week, and half-strength Hoagland solution (Phytotechlabs, Lenexa, KS, USA) was applied once a week. Eight bacterial strains were evaluated in three conditions (no stress, salt stress, and pathogen stress); there were three replicates per treatment, with each treatment constituted by four plants. Bacterial strains were root-inoculated twice with 10^8^ ufc/mL of MgSO_4_ 10 mM solution upon transplant and 10 days after. Three days after the second dose, the plants were stress-challenged.

The plants in the pathogen block were kept in a humidity chamber to ensure stomatal opening, allowing for disease progress for 24 h before the pathogen challenge, when the leaf spot pathogen *Pseudomonas syringae* DC3000 was sprayed on the leaves (10^8^ ufc/mL in MgSO_4_ 10 mM); the controls were mock-inoculated with a MgSO_4_ 10 mM solution. The plants were harvested one week after the challenge, when the photosynthesis and fresh weigh were determined. The disease incidence was recorded by counting the leaves showing necrotic spots, and expressed as the diseased leaves per plant (%), relative to the disease incidence in the controls; using this, a disease protection index was calculated, considering that the controls represent the maximum disease, according to the following formula:$$\mathrm{Disease}\;\mathrm{protection}\;\mathrm{index}=100-\left(\frac{bacteria\;pathogen-bacteria}{Pathogen-control}\times 100\right)$$

In this formula, bacteria pathogen represents diseased plants in the PGPR + pathogen-treated plants, Bacteria represents the PGPR-treated non-challenged plants, Pathogen represents the diseased control plants, and Control represents the non-challenged plants.

Plants in the salt stress block (salinity) were watered with a NaCl solution (3.5 g/L) 3 days after the second PGPR dose. One week after this, the plants were harvested, the photosynthesis and fresh weight were measured, and the plant wilting was registered. Protection is presented as the relative wilting index, comparing the treatments to the control, according to the following formula:$$\mathrm{Relative}\;\mathrm{wilting}\;\mathrm{index}=100-\left(\frac{bacteria\;salt-bacteria}{salt-control}\times 100\right)$$

#### 4.4.2. Study of the Signal Transduction Pathway Involved in Protection

The following three strains were selected: G7, H47, and L44 (Figure 1). In this experiment, 3 strains in the same 3 conditions (no stress, salt stress, and pathogen stress) were assayed. Each treatment had 3 replicates with 7 plants per replicate. Two-week-old pregerminated *A. thaliana* Col 0 seeds were transplanted to 100 mL pots filled with 3:1 (*v*/*v*) peat/sand mixtures and cultivated as described in Section 4.1, following the same experimental set-up. The leaves were sampled 24 h after the stress challenge, powdered in liquid nitrogen, and stored at −80 °C for analysis by qPCR of the marker genes of the Et/JA acid pathway (*LOX2* and *PDF1*) and the SA pathway (*NPR1* and *PR1*). Harvest was performed to evaluate the photosynthesis, fresh weight, and protection to biotic or abiotic stress.

Prior to the RNA extraction, the samples were ground to a fine powder with liquid nitrogen. The total RNA was extracted from each replicate with the PureLink RNA Micro Kit (Invitrogen, Waltham, MA, USA), with the DNAase treatment included. The RNA purity was confirmed using the NanodropTM. A retrotranscription by RT-qPCR was performed using the iScript tm cDNA Synthesis Kit (Bio-Rad, Hercules, CA, USA). All of the retrotranscriptions were carried out using a GeneAmp PCR System 2700 (Applied Biosystems, Foster City, CA, USA) with the following protocol: 5 min at 25 °C, 30 min at 42 °C, 5 min at 85 °C, and holding at 4 °C. Amplification was carried out with a MiniOpticon Real Time PCR System (Bio-Rad, Hercules, CA, USA) with the following protocol: 3 min at 95 C, and then 39 cycles consisting of 15 s at 95 °C, 30 s at 55 °C, and 30 s at 72 °C, followed by a melting curve to check results (the melting curves are shown in Appendix A). To describe the expression obtained in the analysis, the cycle threshold (Ct) was used. Standard curves were calculated for each gene, and the efficiency values ranged between 90 and 110%. The results for the gene expression were expressed as a differential expression by the 2^−∆∆Ct^ method [44]. Previous assays demonstrated the better stability of the *SAND* gene over the *ACTIN* gene; therefore, the *SAND* gene was used as the reference (housekeeping) gene. The *PDF1* (*AtPDF1*), *PR1* (*AtPR1*), *NPR1* (*AtNPR1*), and *LOX2* (*AtLOX2*) genes were studied. The gene primers used are shown in Table 3. The *PDF1*, *PR1*, and *LOX2* primers were from Martin-Rivilla et al. [46], the *NPR1* primer was from Chen et al. [47], while the housekeeping primer (*SAND*) was designed using PRIMER3 (AT2G28390). The differential expression of the treatment with bacteria with respect to its control was studied in conditions of salt stress, pathogen stress, and without stress; when the differential expression was greater than 1 or less than −1, the results were considered significant.

### 4.5. Photosynthesis

The photosynthetic efficiency was determined through the chlorophyll fluorescence emitted by photosystem II. A pulse amplitude-modulated (PAM) fluorometer (Hansatech FM2, Hansatech, Inc., Narborough, UK) was used to measure the chlorophyll fluorescence. After the dark-adaptation of the leaves, a weak modulated irradiation (1 μmol m^−2^ s^−1^) was applied to measure the minimal fluorescence (F_0_; dark-adapted minimum fluorescence). The maximum fluorescence (Fm) was determined from the dark-adapted state by delivering a 700 ms saturating flash (9000 μmol m^−2^ s^−1^). The variable fluorescence (Fv) was calculated as the difference between the maximum fluorescence (Fm) and the minimum fluorescence (F_0_). The maximum photosynthetic efficiency of photosystem II (the maximal PSII quantum yield) was calculated as Fv/Fm. Immediately, the leaf was continuously irradiated with red–blue actinic beams (80 μmol m^−2^ s^−1^) and, after equilibrating for 15 s, the Fs was recorded (steady-state fluorescence signal). Then, another saturation flash (9000 μmol m^−2^ s^−1^) was applied to determine the Fm’ (the maximum fluorescence under light-adapted conditions). Other fluorescent parameters were calculated as follows: the effective PSII quantum yield PSR = (Fm’ − Fs)/Fm’ [48] and the non-photochemical quenching coefficient NPQ = (Fm − Fm’)/Fm’. All of the measurements were carried out in the 6 plants for each treatment.

### 4.6. Statistics

To evaluate the onset of stress on the plants, Student’s *t*-test (*p* < 0.05) was carried out between the controls and the stress controls (non-bacterized) for each variable. In addition, a one-way analysis of variance with replicates was carried out to compare the bacterial effects on all of the parameters under each stress condition. When significant differences were detected (*p* < 0.05), a Fisher test was used. Prior to the analysis, normality was tested by the Shapiro–Wilk test (*p* > 0.05) and the homoscedasticity of the variance was evaluated by Levene’s test (*p* > 0.05). The analyses were performed with the computer program Statgraphics plus 5.1 (Statistical Graphics Corp., Princeton, NJ, USA).

## 5. Conclusions

In summary, (i) the three strains prime *Arabidopsis* according to the more marked and faster increases in the gene expression upon the stress challenge; (ii) all three strains activate the SA-mediated and the Et/Ja-mediated pathways, therefore conferring a wide protection against stress; (iii) *PR1* and *PDF1*, associated to SAR and ISR protection against pathogen stress, respectively, are also overexpressed under abiotic stress conditions. Based on the above data, the priming of the plant adaptive metabolism is strain-dependent, although stress challenge prioritizes the intensity of each pathway. The trans-priming induced by the three PGPR strains evaluated herein confirms these strains as excellent potential candidates for biofertilizers due to their wide protection.

## Figures and Tables

**Figure 1 plants-13-03565-f001:**
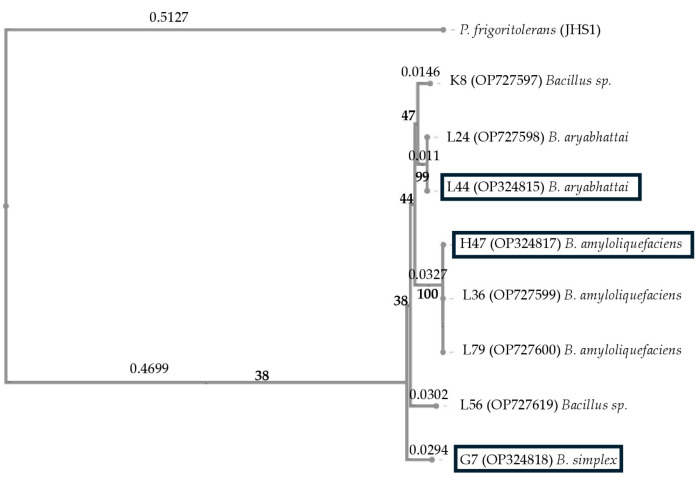
A phylogenetic tree constructed with the 16S rRNA sequences. A rooted phylogenetic tree (the *Peribacillus frigoritolerans* strain JHS1 was used as the outgroup) was constructed with MAFFT v7.0, with the sequences aligned in the same program. Neighbor joining was used to infer the evolutionary distances (the numbers on the branches) with a bootstrap of 1000 replicates. The annotation of the bacteria included in the tree was obtained from the NCBI (https://www.ncbi.nlm.nih.gov/). Selected strains are squared.

**Figure 2 plants-13-03565-f002:**
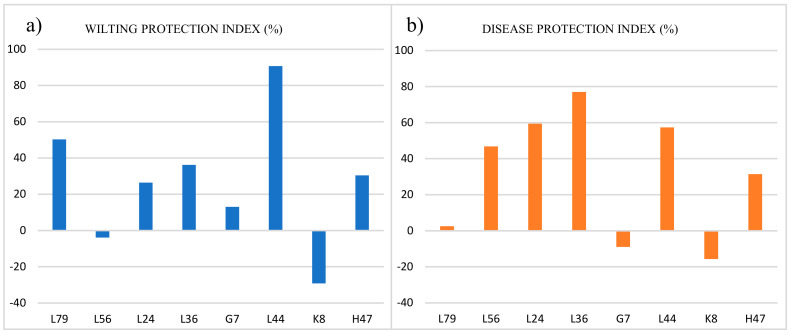
Protection index (%) against (**a**) salt stress and (**b**) leaf pathogen infection in 4-week-old plants inoculated with L79, L56, L24, L36, G7, L44, K8, and H47 one week after the challenge (salt or pathogen). Protection is presented as the relative wilting/disease index relative to the control.

**Figure 3 plants-13-03565-f003:**
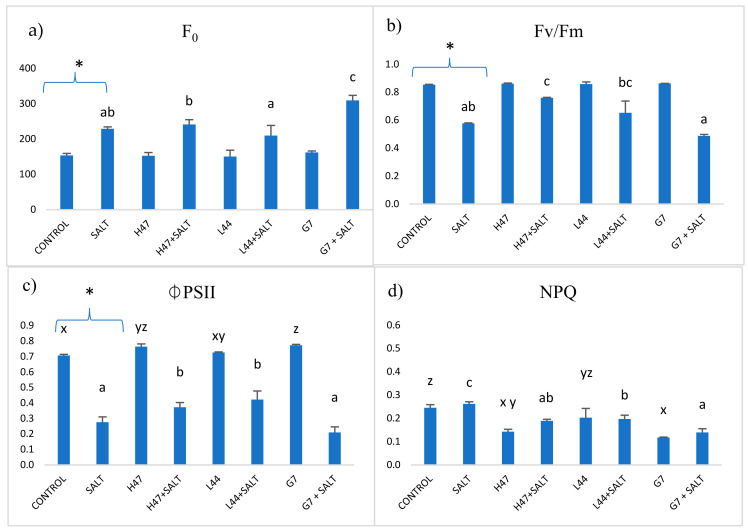
Photosynthetic parameters from the photosystems and the light reactions measured in the control plants and plants inoculated with H47, G7, and L44, which were subjected to salt stress. (**a**) Minimum fluorescence in dark-adapted leaves (F0). (**b**) Maximum potential photosynthetic efficiency of photosystem II (Fv/Fm). (**c**) Photosynthetic efficiency of photosystem II (φPSII) and (**d**) non-photochemical quenching (NPQ). For each treatment and parameter, the mean value and the standard error (n = 6) is represented. The asterisks (*) represent significant differences, according to the Student’s *t*-test (*p* < 0.05), between the control and the control with salt stress. Different letters represent significant differences, according to the ANOVA, between the treatments without salt stress (x,y,z) and the treatments with salt stress (a,b,c) (*p* < 0.05).

**Figure 4 plants-13-03565-f004:**
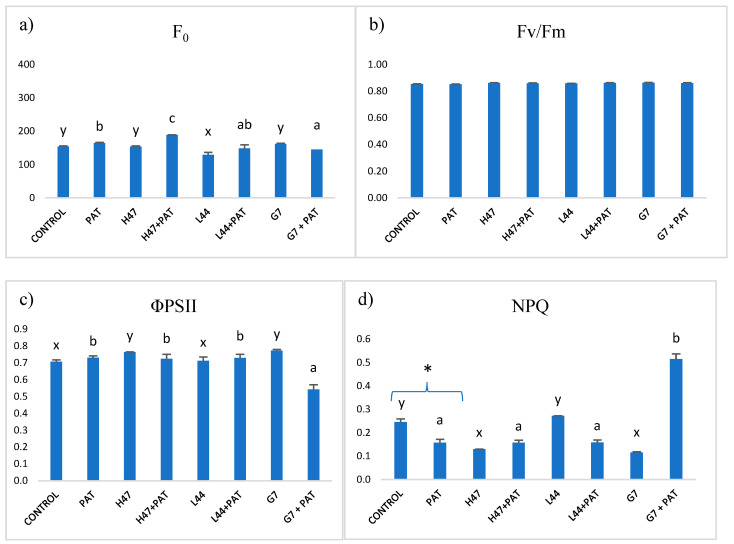
Photosynthetic parameters from the photosystems and the light reactions measured in the control plants and the plants inoculated with H47, G7, and L44, which were subjected to pathogen stress. (**a**) Minimum fluorescence in dark-adapted leaves (F0). (**b**) Maximum potential photosynthetic efficiency of photosystem II (Fv/Fm). (**c**) Photosynthetic efficiency of photosystem II (φPSII) and (**d**) non-photochemical quenching (NPQ). For each treatment and parameter, the mean value and the standard error (n = 6) is represented. The asterisks (*) represent significant differences, according to the Student’s *t*-test (*p* < 0.05), between the control and the control with the pathogen challenge. Different letters represent significant differences, according to the ANOVA, between the treatments without pathogen challenge (x,y,z) and treatments with pathogen challenge (a,b,c) (*p* < 0.05).

**Table 1 plants-13-03565-t001:** Dry weight (mg) in inoculated and non-inoculated plants under salt stress and non-stressed plants. Data are the average ± standard error (n = 12). Asterisks (*) indicate significant differences to the controls according to Student’s *t*-test (*p* < 0.05).

	Dry Weight No Stress (mg)	Dry Weight Salt Stress (mg)
Control	9.87 ± 0.92	6.13 ± 0.79
L24	12.87 ± 1.85	12.53 ± 1.37 *
L44	6.17 ± 0.34 *	7.60 ± 1.37
K8	8.93 ± 0.69	13.20 ± 0.98 *
G7	13.50 ± 1.44 *	11.93 ± 1.2 *
L56	4.10 ± 0.86 *	13.75 ± 4.24
H47	5.70 ± 0.49 *	6.93 ± 0.74
L36	9.65 ± 0.95	15.67 ± 1.60 *
L79	6.17 ± 0.92 *	11.03 ± 1.83

**Table 2 plants-13-03565-t002:** Differential expression of target genes in *A. thaliana* after 2 inoculations with the PGPR relative to the non-bacterized control, 24 h after exposure to the bacterium, to salt stress or to pathogen challenge. Strains: (**A**) H47, (**B**) L44, and (**C**) G7. When the differential expression is greater than 1 or less than −1, the results are significant. The target genes include the *NPR1* gene, and the marker genes of the Ja/Et (*PDF1* and *LOX2*) and the SA pathways (*PR1*).

(A)			
	Control vs. H47	Salt vs. H47-Salt	Pathogen vs. H47-Pathogen
*NPR1*	0.05	−2.01	1.12
*PDF1*	−1.75	1.18	−0.06
*LOX2*	1.52	−1.18	1.78
*PR1*	−0.36	1.09	0.20
**(B)**			
	**Control vs. L44**	**Salt vs. L44-Salt**	**Pathogen vs. L44-Pathogen**
*NPR1*	0.60	−0.74	−0.40
*PDF1*	−2.59	3.00	−1.37
*LOX2*	2.37	−0.40	0.29
*PR1*	0.85	0.77	1.09
**(C)**			
	**Control vs. G7**	**Salt vs. G7-Salt**	**Pathogen vs. G7-Pathogen**
*NPR1*	−1.01	1.72	0.72
*PDF1*	−1.52	−3.53	0.84
*LOX2*	0.76	2.60	0.39
*PR1*	0.36	3.32	2.23

**Table 3 plants-13-03565-t003:** Primers used for the RT-qPCR expression analysis.

Gene Identifier	Gene	Forward Primer	Reverse Primer
*AtPDF1*	*PDF1* [46]	5′-TTGTTCTCTTTGCTGCTTTCGA	5′-TTGGCTTCTCGCACAACTTCT
*AtPR1*	*PR1* [46]	5′-AGTTGTTTGGAGAAAGTCAG	5′-GTTCACATAATTCCCACGA
*AtLOX2*	*LOX2* [46]	5′ACTTGCTCGTCCGGTAATTGG	5′-GTACGGCCTTGCCTGTGAATG
*At NPR1*	*NPR1* [47]	5′-TTTGGAAGGTAGAACCGCAC	5′-ACATTCAACCGCCATAGTGG
*AtSAND*	*SAND* (AT2G28390)	5′-CTGTCTTCTCATCTCTTGTC	5′-TCTTGCAATATGGTTCCTG

## Data Availability

The original contributions presented in the study are included in the article/Appendix A, further inquiries can be directed to the corresponding author.

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
