# Peer review of "The Crossregulation Triggered by *Bacillus* Strains Is Strain-Specific and Improves Adaptation to Biotic and Abiotic Stress in Arabidopsis"

_plants, 2024, doi:10.3390/plants13243565_

Round 1

Reviewer 1 Report

Comments and Suggestions for Authors

Dear authors,

you can find in the file attached all my comments and suggestions given to improve your paper.

With my best regards.

Reviewer 2 Report

Comments and Suggestions for Authors

This article presents an interesting study of the priming mechanisms by different strains of Bacillus against biotic and abiotic stress. The title represents the contentment of the manuscript and the abstract adequately summarizes it. The introduction provides the necessary backroad for the presented research. Results and methods are presented clearly. The conclusions are supported by the results and cited literature. The manuscript however contains a few linguistic mistakes, some of which I pointed out below, however, I recommend that the article should be checked for such mistakes. The citation style does not match the Plant journal requirements. The presented results are properly described in the main text, but additional supporting data should be added to the supplement, e.g. melting curves, statistics, etc. I do regret that the authors did not analyze strains L79, L56, since it could give insight into what signaling pathway is used in priming a biotic and abiotic separately. Additionally, 16S sequencing is considered insufficient for assignment to the species level. This does not negatively influence the content of this manuscript, but if you consider working with these strains in the future, consider sequencing at least one additional gene. The most problematic part is the usage of a single reference gene without information on the stability of its expression. I understand that reference gene candidates are considered relatively stable but a two-fold difference may happen, which would significantly influence the data interpretation. Please add information, on how this reference gene was selected, and how its stability was verified to the supplementary materials. Concluding I recommend this article to be published in Plants after major revisions. My recommendation for major revision is due to the lack of supplementary material which in my opinion must be included in this kind of study, and therefore I would like to revise this manuscript again to confirm that all necessary information is added to the supplement.

Line 13: recruit

Line 14 what potential

Line 17: why challenging

Line 20: Please include full scientific names whenever the species appears for the first time in the title, abstract, main text, and figure or table caption.

Line 33??

Line 38: factors

Line 45: Please change the citation style

Line 229 it is worth what?

Line 306 Lysogenic Broth

Line 357: those formulas are unclear, how the parameters included in the formula were measured

Line 338: You only added one reference gene? Did you check the stability of its expression? If this gene is selected based on previous studies it is of utmost importance to indicate it here, the 2-fold difference can be easily different if another reference gene is selected.

Line 381: Please use the degree symbol or Celsius degree symbol

Comments on the Quality of English Language

 The manuscript however contains a few linguistic mistakes, some of which I pointed out in my comments however, I recommend that the article should be checked for such mistakes. 

Reviewer 3 Report

Comments and Suggestions for Authors

Dear Authors

Your experiment is interesting; you did a lot of work but this all suffers from poor presentation in the article. It definitely needs editing.

My comments:

Title: Although the title «Arabidopsis thaliana Priming by Bacillus Strains Is Strain-Specific and Involves the SA- and Et/Ja- Pathway Conferring Protection against Biotic (Pseudomonas syringae DC3000) and Abiotic Stress (Salinity)» is very informative about the content of the article and in fact the reader knows from the beginning the result of the research, I find it too long with too much information. My suggestion is to enrich the keywords and remove information from the title.

Abstract: L12-17 could be less. Too much introduction of your work

The abstract needs reformation

L21 16S rRNA

L35 plant pathogen

Introduction

L41 UN: written in full

References in ”Plants” have to be numbers in brackets [1],

English language needs editing

Results: put titles and divide your results into paragraphs. Not to be all in one package.

Figure 1. Enrich your Phylogenetic tree with more Bacillus species and strains to make it more interesting.

The results of the study are interesting and sufficient to support publication in Plants but are poorly or not at all organized. They should be separated and grouped with headings by paragraph.

M&M L303-302  something is missing

L315 16S rRNA genes

L315-316 that set of primers has its own references

L333 you should use the current MEGA version

Comments on the Quality of English Language

English language needs editing by a native speaker

Round 2

Reviewer 2 Report

Comments and Suggestions for Authors

I would like to thank the authors for updating their manuscript according to my suggestions. I would recommend adding melting curves for the qPCRs to prove the effectiveness of selected primers and add the information about Actin stability. I the stability of  SAND and actin was checked it would show that initially there were at lest two reference gene, from which SAND was selected for better stability. Please bear in mind that  the reported 2 fold changes can be attributed to the selection of housekeeping genes, therefore it is crucial to confirm their stability. Concluding based on the current version of the manuscript I recommend it to be published after minor revisions.

Line 142: Please expand the captions (they should contain all necessary information to read each graph or table. 

Reviewer 3 Report

Comments and Suggestions for Authors

Dear Authors

The article plants-3252229_R1 “Arabidopsis thaliana Priming by Bacillus Strains Is Strain-Specific and Involves the SA- and Et/Ja- Pathway Conferring Protection against Biotic and Abiotic Stress” appears this time much improved compared to the original.

The English language seems improved as well as the presentation of the description of the experiments and the results.

L19, 113, 146 etc: The 16S has to be written by capital S

Comments on the Quality of English Language

The English language of the text has been improved

Author Response

PLEASE SEE THE ATTACHEMENT
